# Clinical and biomarker results from a phase II trial of combined cabozantinib and durvalumab in patients with chemotherapy-refractory colorectal cancer (CRC): CAMILLA CRC cohort

Anwaar Saeed [1,2] ✉, Robin Park[3], Harsh Pathak[4], Ayah Nedal Al-Bzour [1], Junqiang Dai[5], Milind Phadnis[5], Raed Al-Rajabi[6,7], Anup Kasi[6,7], Joaquina Baranda [6,7], Weijing Sun [6,7], Stephen Williamson[6,7], Yu-Chiao Chiu [2], Hatice Ulku Osmanbeyoglu [2], Rashna Madan[4], Hassan Abushukair [1], Kelly Mulvaney[7], Andrew K. Godwin [4,7,8] & Azhar Saeed[9]

CAMILLA is a basket trial (NCT03539822) evaluating cabozantinib plus the ICI durvalumab in chemorefractory gastrointestinal cancer. Herein, are the phase II colorectal cohort results. 29 patients were evaluable. 100% had confirmed pMMR/MSS tumors. Primary endpoint was met with ORR of 27.6% (95% CI 12.7-47.2%). Secondary endpoints of 4-month PFS rate was 44.83% (95% CI 26.5-64.3%); and median OS was 9.1 months (95% CI 5.8-20.2). Grade≥3 TRAE occurred in 39%. In post-hoc analysis of patients with RAS wild type tumors, ORR was 50% and median PFS and OS were 6.3 and 21.5 months respectively. Exploratory spatial transcriptomic profiling of pretreatment tumors showed upregulation of VEGF and MET signaling, increased extracellular matrix activity and preexisting anti-tumor immune responses coexisting with immune suppressive features like T cell migration barriers in responders versus non-responders. Cabozantinib plus durvalumab demonstrated anti-tumor activity, manageable toxicity, and have led to the activation of the phase III STELLAR-303 trial.

Colorectal cancer (CRC) is the 4th most common cause of new cancer diagnosis and the 2nd leading cause of cancer-related mortality in the United States[1]. Although the clinical outcomes of CRC have recently improved with 5-year relative survival rates of ~65% for all stages combined, advanced or metastatic CRC has a poor prognosis with 5-year OS of ~15%[1]. The standard of care (SOC) frontline therapies for metastatic CRC are chemotherapy regimens that include fluorouracil in combination with platinum, irinotecan, or both, depending on patient comorbidities and functional status. In a small subset of patients (3–5%) with metastatic CRC whose tumors are mismatch

[1]Department of Medicine, Division of Hematology & Oncology, University of Pittsburgh Medical Center (UPMC), Pittsburgh, PA, USA. [2]UPMC Hillman Cancer Center, Pittsburgh, PA, USA. [3]Division of Hematology and Medical Oncology, Moffitt Cancer Cente, Tampa, FL, USA. [4]Department of Pathology and Laboratory Medicine, University of Kansas Medical Center, Kansas City, KS, USA. [5]Department of Biostatistics, University of Kansas Medical Center, Kansas City, KS, USA. [6]Department of Medicine, Division of Medical Oncology, University of Kansas Medical Center, Kansas City, Ks, USA. [7]University of Kansas Cancer Center, Kansas City, KS, USA. [8]Kansas Institute for Precision Medicine, University of Kansas Medical Center, Kansas City, KS, USA. [9]Department of Pathology and Laboratory Medicine, University of Vermont Medical Center, Burlington, VT, USA. ✉e-mail: saeeda3@upmc.edu

repair deficient or microsatellite instability-high (dMMR/MSI-H), immune checkpoint inhibitor (ICIs) with anti-programmed death-1 (PD-1) inhibitors with or without anti-CTLA-4 inhibitors have excellent efficacy with response rates up to 65% and median progression-free survival (mPFS) longer than 16.5 months and are the recommended frontline therapies. The favorable response to ICIs is due to the inflamed tumor microenvironment (TME) and increased tumor mutational burden of dMMR/MSI-H tumors[2,3]. However, the tumors of most patients (>95%) with metastatic CRC are proficient MMR/microsatellite stable (pMMR/MSS), have features of TME that are not associated with cytotoxic anti-tumor immune responses, and consequently do not respond to ICIs. After progression on first- and second-line chemotherapy including epidermal growth factor receptor (EGFR) monoclonal antibodies in the *RAS* wild-type population and in patients with pMMR/MSS tumors who do not harbor targetable driver molecular alterations, salvage therapy options are limited to regorafenib (multi-tyrosine kinase inhibitor) and tipiracil/trifluridine (TAS-102) with or without bevacizumab, which have suboptimal efficacy and toxicity[4–6]. Thus, the development of novel treatment strategies, especially aimed at modulating the TME to a state that is associated with cytotoxic anti-tumor immune responses with the goal of potentiating ICI in pMMR/MSS metastatic CRC patients is an area of great unmet clinical need.

Cabozantinib is an oral multi-target tyrosine kinase inhibitor (TKI) with target affinities against the receptor tyrosine kinases (RTKs) vascular endothelial growth factor receptor (VEGFR)−2, mesenchymal epithelial transition factor (MET)/hepatocyte growth factor (HGF), AXL, MER, and TYRO3. It has immunomodulatory properties that counteract the immune suppressive TME and stimulate local and systemic anti-tumor immune responses for various advanced solid tumors[7–11]. Pre-clinical studies show cabozantinib has synergistic anti-tumor activity with ICIs in patient-derived xenograft models of pMMR/MSS CRC[12]. In a phase II clinical trial, cabozantinib has demonstrated single-agent disease-stabilizing activity in patients with chemorefractory pMMR/MSS metastatic CRC[13]. Additionally, cabozantinib combined with ICIs such as with atezolizumab (anti-PD-L1), nivolumab (anti-PD-1) with or without ipilimumab (anti-CTLA-4), or pembrolizumab (anti-PD-1) have demonstrated anti-tumor activity in early phase trials in patients with advanced genitourinary, hepatocellular, and non-small cell lung carcinoma[14–20]. Thus, cabozantinib plus ICI represents a promising avenue for immune-potentiating strategy in patients with pMMR/MSS metastatic CRC who otherwise do not respond to single-agent ICIs.

CAMILLA is an ongoing phase I/II trial that is evaluating cabozantinib plus durvalumab in patients with advanced, previously treated gastrointestinal cancer. The completed phase I part of the study has been published and has established the recommended phase II dose and highlighted the manageable toxicity of the combination regimen as well as early encouraging signal of efficacy in patients with metastatic or unresectable CRC[21]. The phase II part of the study is currently ongoing and consists of four patient cohorts including two cohorts with hepatocellular carcinoma; one with gastric, gastroesophageal junction, or esophageal adenocarcinoma; and one with CRC. Herein, we present the final phase II CRC cohort results of CAMILLA, in which we have evaluated cabozantinib plus durvalumab in patients with pMMR/MSS advanced CRC who have progressed on previous SOC systemic therapy.

## Results

### Patients

Thirty-one patients were enrolled in the study. Among them, 29 patients were evaluable for efficacy (Table 1). Two patients were not evaluable for efficacy as they did not receive at least 1 month of trial regimen and without evidence of disease progression (Supplementary

**Table 1 | Baseline participant characteristics**

| Characteristic | All participants (*N* = 29) |
|---|---|
| Age, years | |
| Median (range) | 57 (27–76) |
| 60 and older | 13 (45) |
| Males | 13 (45) |
| ECOG performance status | |
| 0 | 3 (10) |
| 1 | 26 (90) |
| Sidedness | |
| Left sided | 25 (86) |
| Rectum | 16 (55) |
| Right sided | 4 (14) |
| *RAS* gene status | |
| Mutant | 17 (59) |
| Wild | 12 (41) |
| *BRAF* gene status | |
| Mutant | 0 (0) |
| Wild | 29 (100) |
| *HER2* amplification | 2 (6.9) |
| MSI/MMR status | |
| MSI-H/dMMR | 0 (0) |
| MSS/pMMR | 29 (100) |
| Prior lines of therapy | |
| Median | 3 |
| 2 | 14 (48) |
| ≥ 3 | 15 (52) |
| Prior therapies | |
| Trifluridine/tipiracil | 4 (14) |
| Regorafenib | 0 (0) |
| Bevacizumab | 22 (76) |
| Anti-EGFR | 14 (48) |
| Number of metastatic sites | |
| <3 | 0 (0) |
| ≥3 | 29 (100) |
| Patients with Liver metastasis | 23 (79) |

Data area presented as No. (%) unless otherwise noted.
*ECOG* Eastern Cooperative Oncology Group, *MSI-H/dMMR* microsatellite instability-high or mismatch repair deficient, *MSS/pMMR* microsatellite stable or mismatch repair proficient.

Fig. 9). Median age was 57 (range 27–76). All patients had ≥ three metastatic sites and had progressed on two or more prior lines of systemic therapy. Median lines of prior therapy were three (range 2–4). Fifteen patients (52%) had received more than three prior lines of therapy. Most of the patients had left-sided tumors (86%). Liver metastases were present in 23 out of 29 patients (79%). Hundred percent of the enrolled patients had pMMR/MSS tumors. Twelve out of 29 patients (41%) had *RAS* wild-type tumors and zero out of 29 (0%) had pathogenic *BRAF V600E* mutation. *HER2* amplification was identified in two out of twenty-nine patients (6.9%). Most patients had received prior bevacizumab (76%), approximately half had received prior epidermal growth factor (EGFR) antibodies (48%), and a few (13%) had received prior tipiracil/trifluridine. No patients had received prior regorafenib.

### Efficacy

Overall, 8 out of 29 patients achieved an objective response (objective response rate (ORR) 27.6%, 95% CI 12.73–47.2%) and 25 out of 29 patients achieved disease control (DCR 86.2%, 95% CI 68.3–96.1%) (Figs. 1 and 2, Supplementary Fig. 10, Table 2). Additionally, median

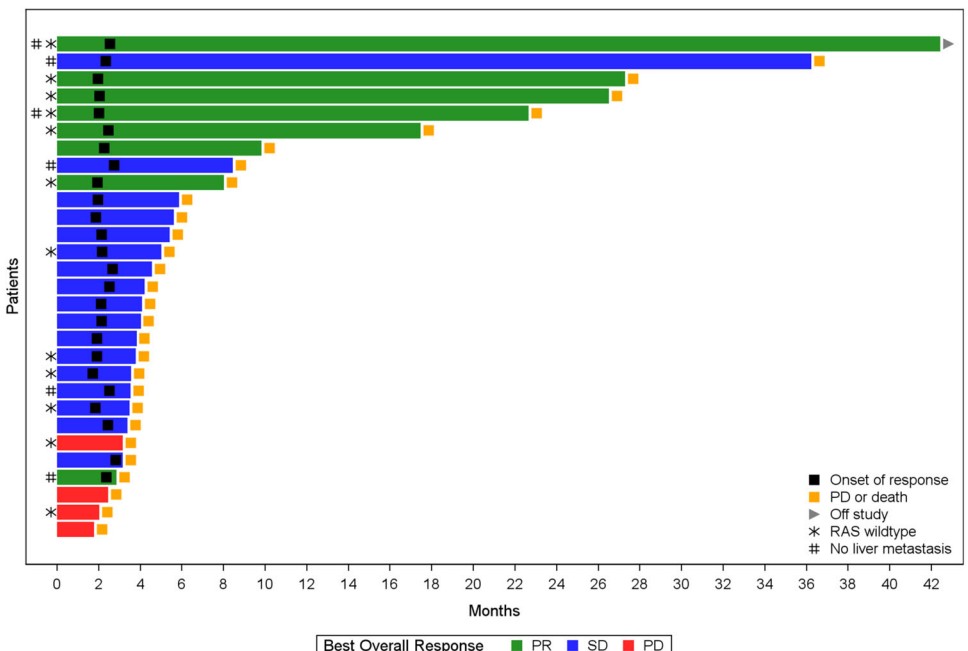

**Fig. 1 | Swimmer plot of individual patient responses.** Colors of bars represent best overall response (green, partial response; blue, stable disease; red, progressive disease). Black filled square marks the date of response onset; yellow filled square represents the date of progression or death; gray filled arrow denotes that the patient is currently active on study; star denotes a *RAS* gene wild-type patient; and # represents a patient with no liver metastasis.

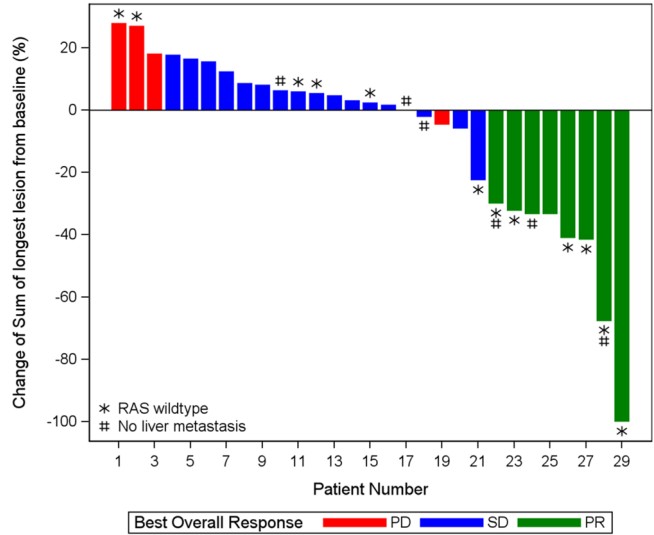

**Fig. 2 | Waterfall plot of individual patient responses.** Colors of bars represent best overall response (green, partial response; blue, stable disease; red, progressive disease). A star denotes a *RAS* wild-type patient and # represents a patient with no liver metastasis.

progression-free survival (PFS) was 3.7 (95% CI 3.4–5.7 months) and median overall survival (OS) was 9.1 months (95% CI 5.8–20.2 months) (Supplementary Figs. 11 and 12). Thirteen out of 29 patients (44.8%) and 9 out of 29 patients (31%) were progression-free at 4 months and 6 months, respectively. In a post-hoc subgroup analysis, of 12 patients with *RAS* wild-type tumors, 6 had objective response and 10 had disease control. The median PFS was 6.3 months (95% CI 1.8–22.5 months) and median OS was 21.5 months (95% CI 4.5 months not estimable) (Supplementary Table S1, Supplementary Figs. 2 and 3). Among patients with confirmed partial response, all (6/6) had *RAS* wild-type tumors and 4 of those patients had liver metastasis (Supplementary Table 3). Per available comprehensive tumor next-generation sequencing (NGS) reports, done as part of SOC practices, we've looked at known predictive markers of ICIs response such as tumor mutational burden and *POLE/POLD* mutations. None of our responders had *POLE/POLD* mutations or high tumor mutational burden.

## Safety

Thirty-one patients were evaluable for safety. Grade three or higher treatment related adverse events (TRAE) were observed in 39% of patients (Table 3 and Supplementary Data 7). Severe immune-related adverse events (irAEs) were observed in six out of thirty-one (19%) patients. Dose interruptions were required in ten patients for durvalumab and dose interruptions or modifications were required in fourteen patients for cabozantinib respectively. Treatment discontinuation due to TRAE was necessary in three patients for durvalumab and three for cabozantinib respectively. The most common

### Table 2 | Summary of efficacy data

| Variable | All participants (*N* = 29) |
|---|---|
| Overall objective responses (ORR, 95% CI) | 8 (27.6%, 95% CI 12.73–47.2%) |
| Confirmed partial response (ORR, 95% CI) | 6 (20.7%, 95% CI 8–39.7%) |
| Best overall response | |
| Complete response | 0 |
| Partial response | 8 (27.6%) |
| Stable disease | 17 (58.6%) |
| Progressive disease | 4 (13.8%) |
| Disease control rate | 25/29 (86.2%, 95% CI 68.3–96.1%) |
| Median progression-free survival | 3.7 months (95% CI 3.4–5.7 months) |
| Median overall survival | 9.1 months (95% CI 5.8–20.2 months) |
| 4-month PFS rate | 13/29 (44.8%, 95% CI 26.5–64.3%) |

Data area presented as No. (%, lower and upper limits of 95% confidence interval) unless otherwise noted.
*PFS* progression-free survival, *CI* confidence interval.

grade 1/2 TRAEs were fatigue (65%), transaminitis (58%), nausea (58%), hyperthyroidism (48%), diarrhea (42%), anorexia (42%), and palmar-plantar erythrodysesthesia (35%). The most common grade 3 or higher TRAEs were transaminitis (13%), fatigue (6%), and proteinuria (6%).

### Targeted spatial transcriptomic profiling

Tumor biopsies were obtained for testing from the 29 patients enrolled in the study. Those samples were obtained from various metastatic biopsy sites during the 28-day screening period before initiating the

trial regimen of cabozantinib plus durvalumab. Biopsy sites included liver, lung, adnexa, lymph node and retroperitoneum. Twenty out of 29 patients had sufficient tumor tissue upon evaluation of their H&E sections. Those samples were processed and analyzed for this study as described under the methods section (Supplementary Table S2; clinical and molecular baseline characteristics and tumor biopsy sites of patient's samples included in the DSP study). To identify spatial transcriptomic (ST) correlates associated with durable response we assessed the transcriptomic profile of tumor epithelial compartment (TEC) and the stroma compartment (SC) reflective of the TME in responders vs non-responders (Fig. 3A). Treatment response was defined based on progression-free time of more than 180 days combined with partial or complete tumor treatment response. Of the 20 patients, 4 patients met those predefined criteria and were identified as responders. These four responders are all characterized by *RAS* wild-type molecular profile. Given the tumor response enrichment in the *RAS* wild-type patients subgroup, additional analysis was performed comparing *RAS* wild-type responders versus *RAS* wild-type non-responders.

### Transcriptomic characteristics associated with tumor and stroma in responders

We performed differential expression analysis to identify genes that are associated with treatment response in the TEC or SC (Fig. 3B–E; full results in Supplementary Data 1). In the TEC, responders showed significant upregulation of genes encoding extracellular matrix (ECM) proteins including fibronectin (*FN1*) and osteopontin (*SPP1*). Genes related to angiogenesis were significantly upregulated in responders, such as vascular endothelial growth factor A encoding gene (*VEGFA*) in

**Table 3 | Summary of safety data**

| Parameter, *n* (%) | Cabozantib + Durvalumab Overall (*n* = 31) |
|---|---|
| TRAEs ≥ grade 3, number of patients (%) | 12 (39%) |
| Grade 3 | 10 (32%) |
| Immunotherapy-related events | 5 (16%) |
| Grade 4 | 2 (6%) |
| Immunotherapy-related events | 1 (3%) |
| Grade 5 | 0 (0%) |
| Dose modifications (number of patients)<br>• Durvalumab dose interruptions due to TRAEs<br>• Cabozantinib dose interruptions or modifications due to TRAEs<br>• Discontinuation of cabozantinib or durvalumab due to TRAEs | 10 – Durvalumab Treatment delay/hold<br>14 – Cabozantinib Treatment hold/modifications<br>3 – Durvalumab Treatment Discontinuation<br>3 – Cabozantinib Treatment Discontinuation |

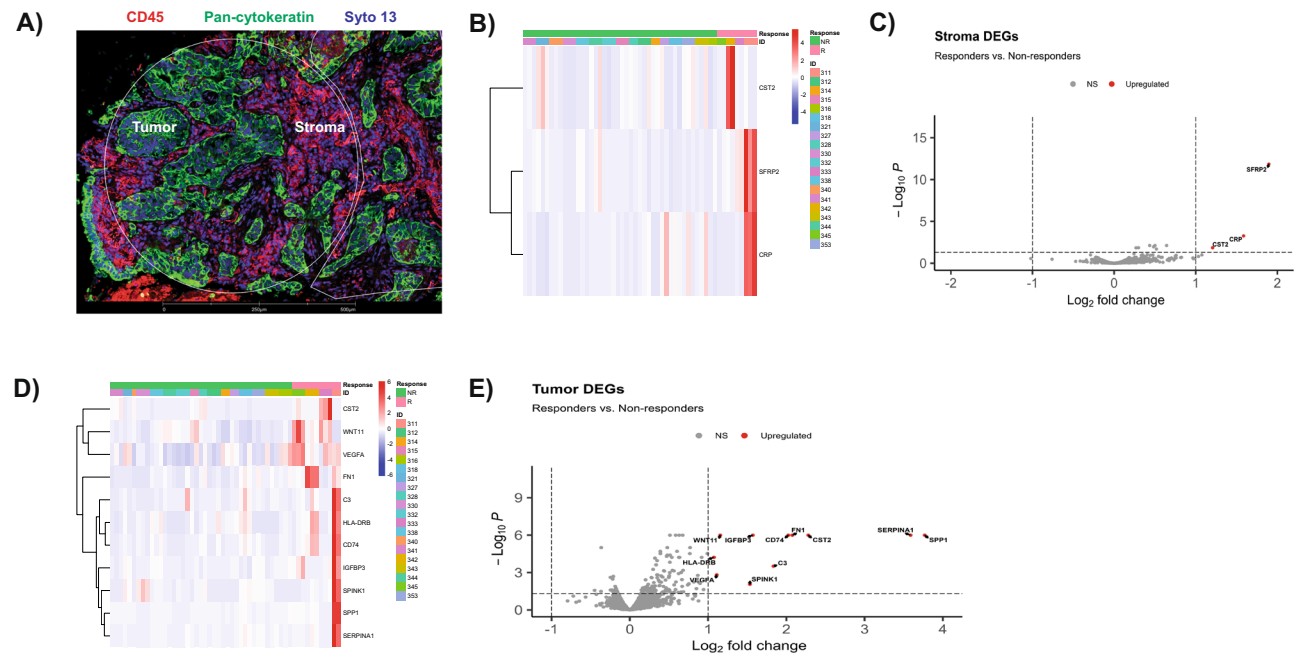

**Fig. 3 | Regions of interest definition and GeoMx spatial transcriptomic analysis results in responders versus non-responders. A** Representation of immunofluorescent staining of one region of interest. Pan cytokeratin (green) was used to identify tumor epithelium for selection of the tumor epithelial compartment. Syto-13 (blue) was used to label nuclei and CD45 (red) was used to label leukocytes. Negative selection approach was used to identify the stroma compartment reflective of the tumor microenvironment that includes CD45 positive and pan-cytokeratin negative cells. **B** Heatmap showing the scaled Q3-normalized expression of the differentially expressed genes (DEGs) between responders (R) and non-responders (NR) in stroma compartment. Patients are depicted by key (ID). **C** Volcano plot for the DEGs in R versus NR in stroma compartment. Significant

DEGs are labeled and shown in red, with log$_2$FC > 1 and adjusted *p*-value < 0.05. NS non-significant. **D** Heatmap showing the scaled Q3-normalized expression of the DEGs between R and NR in tumor epithelial compartment. Patients are depicted by key (ID). **E** Volcano plot for the DEGs in R versus NR in tumor epithelial compartment. Significant DEGs are labeled and shown in red, with log$_2$FC > 1 and adjusted *p*-value < 0.05. NS non-significant. The scaled bar (color key) in the heatmaps represents the scaled Q3-normalized expression of the identified DEGs. Volcano plots used generalized linear models to identify the differentially expressed genes between AOIs and *p*-values were corrected for multiple comparison using Benjamini and Hochberg method. All experiments have been repeated twice to ensure reproducibility.

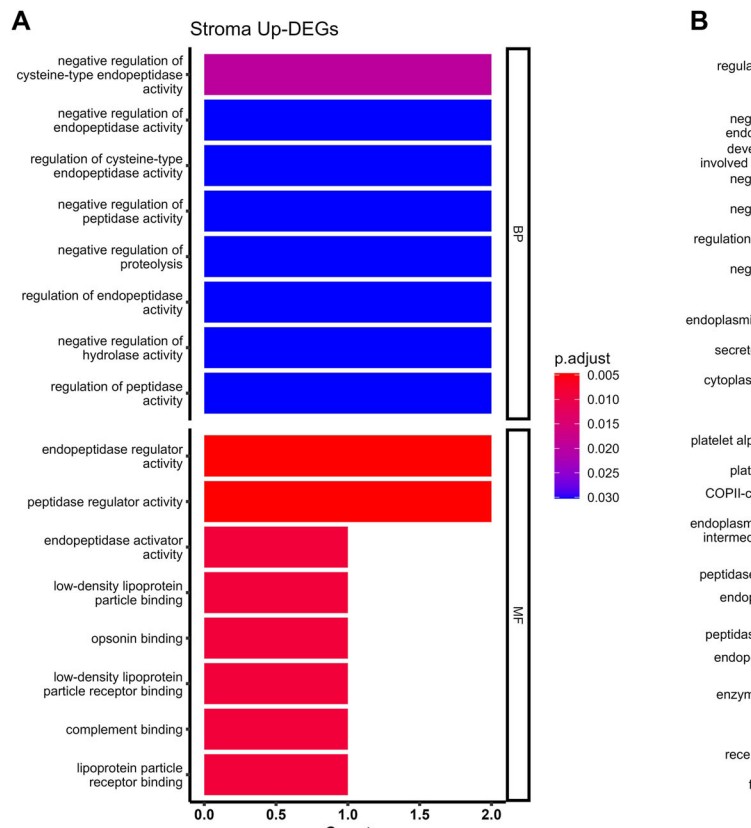

**Fig. 4 | Gene ontology analysis results in responders (R) versus non-responders (NR). A** Bar plots of selected significantly enriched gene ontology (GO) terms including enriched biological processes (BP), cellular components (CC) and molecular function (MF) in R versus NR in stroma compartment. **B** Bar plots of selected significant GO terms including BP, CC and MF, in R versus NR tumor epithelial compartment. P-values were corrected for multiple comparison using Benjamini and Hochberg method.

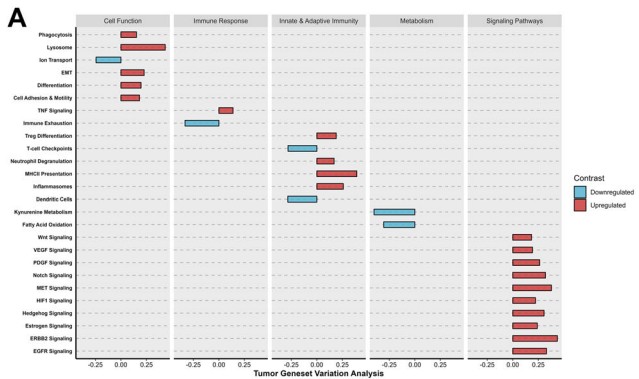

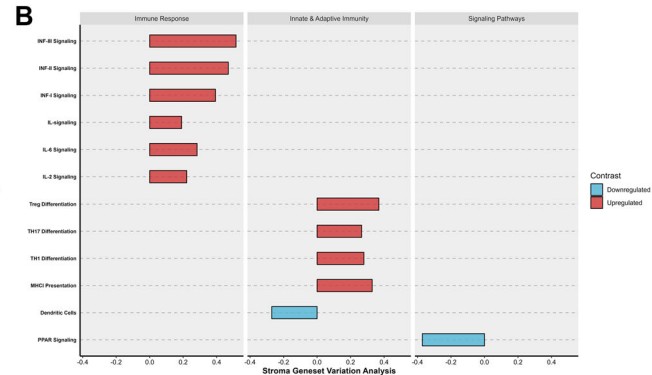

**Fig. 5 | Gene Set Variation Analysis (GSVA) of significant differentially enriched pathways in responders versus non-responders.** The y-axis represents annotated gene sets from the Nanostring Cancer Transcriptome Atlas. The pathways are organized within modules of Cell Function, Metabolism, Immune Response, Innate and Adaptive Immunity and Signaling Pathways. The x-axis represents the fold change difference of differentially enriched pathways in responders in comparison to non-responders. Upregulated pathways are tinted in red and downregulated pathways are tinted in blue. **A** GSVA of tumor epithelial compartment showing significant differentially enriched pathways in responders. **B** GSVA of stroma compartment showing significant differentially enriched pathways in responders. Comparison between responders and non-responders was performed using a linear fit model from Limma with p-values corrected for multiple comparisons using BH method.

TEC and *SFRP2* in SC. *SFRP2* is a Wnt pathway and angiogenesis modulator gene[22]. Functional enrichment analyses of GO (Fig. 4A, B; full results in Supplementary Data 1) and GSVA (Fig. 5A, B; full results in Supplementary Data 2 and 3) revealed concordant results with these observations, showing significant enrichment in signatures related to cell adhesion and motility, fibronectin binding, as well as upregulation of VEGF signaling and HIF1 hypoxia signaling pathway. Upregulation of cell adhesion signaling pathway triggered by cell interaction with ECM was noted in responders including protein tyrosine kinase (PTK2) signaling, a member of focal adhesion kinase (FAK) subfamily (Supplementary Data 3; Reactome GSVA)[23]. Moreover, RTK signaling pathways including MET, EGFR, PDGF and ERBB2 were upregulated in TEC. Notably, interrogation of the Reactome curated database showed upregulation of additional MET signaling pathway related signatures in

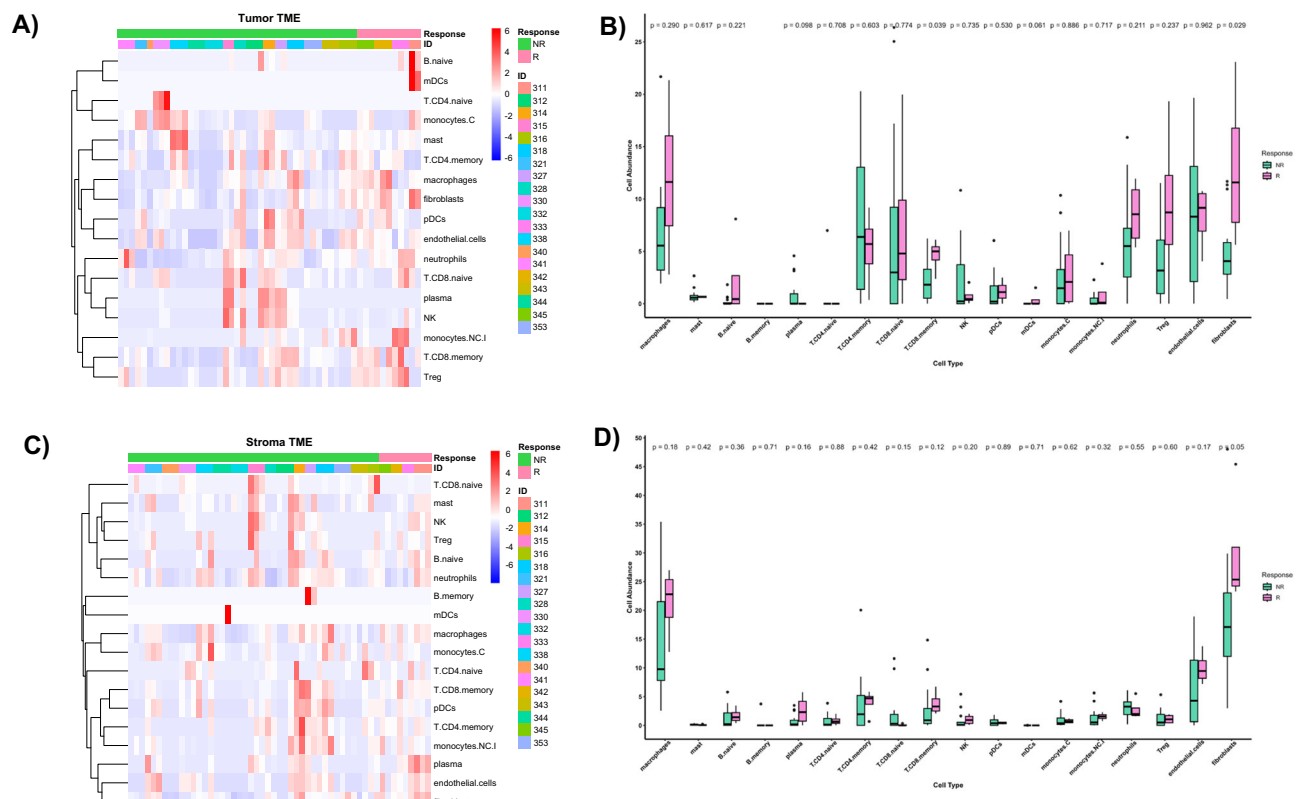

**Fig. 6 | Cell type deconvolution by spatialDecon in responders versus non-responders. A** Heatmap of scaled cell abundance scores (scaled Beta values) in the tumor epithelial compartment. Patients are depicted by key (ID). Plasmacytoid Dendritic cells (pDCs), myeloid Dendritic cells (mDCs), conventional monocytes (monocytes. **C**, non-conventional/intermediate monocytes (monocytes NC.I). **B** Boxplots showing differences of cell infiltration between responders (R, $n = 4$) and non-responders (NR, $n = 16$) in tumor epithelial compartment. Statistical significance was tested using Wilcoxon's rank sum test. **C** Heatmap of scaled cell abundance scores (scaled Beta values) in stroma compartment. Patients are depicted by Key (ID). **D** Boxplots showing differences of cell infiltration between responders (R, $n = 4$) and non-responders (NR, $n = 16$) in stroma compartment. Statistical significance was tested using Wilcoxon's rank sum test. Box and whisker plots show all data points with median as center line with 25th and 75th percentiles. Two-sided Wilcoxon (Mann–Whitney $U$) test was performed.

responders, including MET activates RAS signaling and MET promotes cell motility (Supplementary Data 3). Epithelial mesenchymal transition (EMT) signatures were noted in responders coupled with upregulation of signaling pathways involved in EMT including RTK, Wnt and Notch[24–27]. Of interest, EMT phenotype correlates with the cancer cell ability to adhere to ECM components to promote cell migration[28]. Cell deconvolution analysis showed significant increase in the abundance of fibroblasts in the TEC of responders with marginally significant increase in the fibroblasts in the SC ($p = 0.05$). This finding in combination with upregulation of genes encoding ECM proteins, is suggestive of a dense fibrotic stroma in responders (Fig. 6A–D; Supplementary Data 4) The comparative analysis of responders versus non-responders in the *RAS* wild-type subgroup revealed transcriptomic features in line with the above analysis. In the tumor compartment, responders showed significant upregulation of genes encoding matrix metalloproteinase (MMP7) and FN1 proteins. Genes related to angiogenesis were significantly upregulated in responders including *VEGFA* in TEC and *SFRP2* in SC (Supplementary Fig. 4A–D; full results in Supplementary Data 5). Functional enrichment analyses of GO (Supplementary Fig. 5A, B; full results in Supplementary Data 6) and GSVA (Supplementary Fig. 6A, B; full results in Supplementary Data 6) revealed significant upregulation of MET signaling, enrichment of cell adhesion signatures and fibronectin binding. Cell deconvolution analysis showed a significant increase in the abundance of fibroblasts in the SC of responders (Supplementary Fig. 7A–D).

Collectively, in responders, these findings are suggestive of a possible dense fibrotic tumor stroma with increased ECM activity and angiogenesis. Our data may also indicates active tumor

mechanotransduction, which is the ability of tumor cells to respond to mechanical external stimuli by activating intracellular signaling pathways, as observed by upregulated ECM protein encoding genes, cell adhesion and motility signatures coupled with upregulation of signaling pathways involved in cell adhesion (*i.e*, PTK2) and pathways influencing proliferation and survival (*i.e.*, RTK).

## Immune microenvironment and immune response characteristics of responders

Immune cell deconvolution analysis showed no significant difference in the abundance of CD8+ T cells in the SC. However, CD8+ memory T-cells were significantly more abundant in the TEC of responders in comparison to non-responders (Fig. 6A–D; Supplementary Data 4). Additionally, the responders showed significant enrichment in T-cell inflamed gene expression signature (TIS) in SC (Fig. 7A, B). The TIS was studied and validated by Ayers et al. in multiple cancer types including CRC as a predictive signature for the clinical benefit of PD-1/PDL-1 targeted agents[29]. The enrichment in TIS signifies the presence of preexisting but suppressed T-cell adaptive immune response, IFNγ signaling as well as an inflamed TME. GSVA analysis (Fig. 5A, B; full results in Supplementary Data 2) was in line with this finding and showed upregulation of IFN signaling, T helper cell (TH1) differentiation signature, MHC class I antigen presentation signature in the SC and MHC class II antigen presentation signature in TEC of responders. Upregulation of T regulatory cells (Tregs) differentiation signature in the TEC and SC of responders was noted. No significant difference was found in the immune exhaustion or T-cell counter inhibitory signatures in the SC between responders and non-responders, however

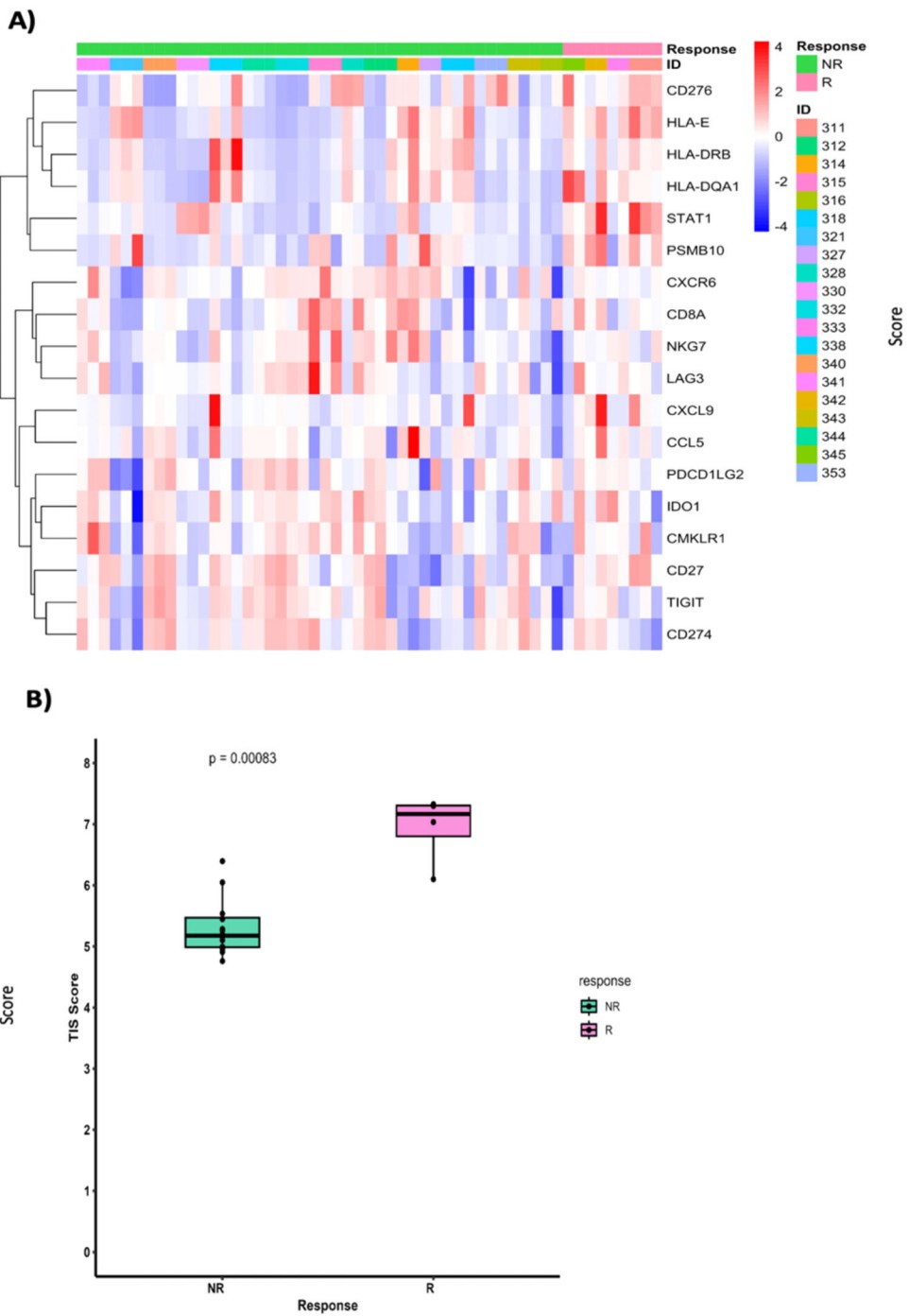

**Fig. 7 | T-cell inflamed gene expression signature (TIS) in responders versus non-responders. A** Heatmap of genes in T-cell inflamed gene expression signature representing the scaled expression of TIS genes. Patients are depicted by key (ID). **B** Boxplots showing differences in the TIS scores between responders and non-responders. The y-axis represents the TIS score. Statistical significance was tested using Wilcoxon's rank sum test. Responders (R, $n = 4$), non-responders (NR, $n = 16$). Box and whisker plots show all data points with median as center line with 25th and 75th percentiles. Two-sided Wilcoxon (Mann–Whitney $U$) test was performed.

responders showed downregulation of immune exhaustion, T-cell check points and tryptophan and kynurenine metabolism signatures in TEC. Additionally, there was a trend toward myeloid dendritic cells exclusion from the TEC and SC in both responders and non-responders. However, no evidence of exclusion was noted for plasmacytoid dendritic cells (Fig. 6A–D). This observation is in line with other published studies that showed lower infiltration of dendritic cells in metastatic sites compared to primary sites in CRC[30,31]. The results of the comparative analysis of responders versus non-responders in the *RAS* wild-type subgroup showed similar findings to the above analysis

including the significant enrichment in the TIS signature in responders (Supplementary Figs. 6A, B, 7A–D, and 8A, B).

Overall, the responders showed evidence suggestive of inflamed TME phenotype with features potentially indicative of preexisting anti-tumor immune responses in association with immunosuppressive characteristics involving Tregs.

## Discussion

In this study, cabozantinib plus durvalumab demonstrated meaningful anti-tumor activity and manageable toxicity in patients with previously

treated pMMR/MSS metastatic CRC, among whom most had liver metastases. Post-hoc subgroup analysis by *RAS* wild-type status was associated with favorable response and survival outcomes. Additionally, increased VEGF and MET signaling as well as cell adhesion and ECM activity signatures were found in responders versus non-responders.

Several early phase trials had failed to identify an active combination ICI regimen for metastatic pMMR/MSS CRC[32–35]. While the original phase I/II REGONIVO study demonstrated a promising activity in patients with metastatic pMMR/MSS CRC treated with regorafenib (multi-TKI) plus nivolumab, the success was not replicated[36]. Subsequent studies showed that in the North American population, combined regorafenib and anti-PD-1/L1 inhibitors had minimal activity (ORR 0–10%, mPFS 1.8–4.3 months) potentially owing to differences in baseline patient characteristics such as degree of tumor burden, performance status, and presence of liver metastases[37–41]. Other clinical studies combining ICI and TKI regimens similarly demonstrated no efficacy including lenvatinib plus pembrolizumab (LEAP-017), which failed to meet the primary endpoint of OS in a phase III trial per a recent press release[42–45]. To date, one of the largest and earliest trials that evaluated a TKI plus ICI combination was the phase III randomized Imblaze370 trial which compared cobimetinib, a MEK inhibitor, plus atezolizumab to regorafenib in refractory pMMR/MSS mCRC[42]. This study notably demonstrated no survival benefit or superiority of cobimetinib plus atezolizumab over regorafenib. The result of this study suggests that specific inhibition of a single pathway is likely to be ineffective in immune modulating the TME and that simultaneous inhibition of more than one signaling pathway is likely needed given the multiple layers of immune evasion in effect in the TME. To this end, cabozantinib is a multi-kinase inhibitor with single-agent disease-stabilizing activity for pMMR/MSS mCRC and immune modulatory effects demonstrated in pre-clinical and clinical settings for various advanced solid tumors[13,46,47]. Thus, our study adds to the growing evidence of the potential immune activation by cabozantinib in advanced immune-cold tumors.

Recent early phase trials have shown signals of efficacy albeit with important limitations. While regorafenib plus nivolumab and ipilimumab (RIN trial) elicited an objective response in 8 out of 29 patients (27.6%), 75% (22/29) of the patients had no liver metastases and responses were restricted to patients without liver metastases. Similarly, the activity of combined balstilimab (anti-PD-1) and bosentilimab (Fc-enhanced anti-CTLA-4) was restricted to patients without liver metastases[48]. Although the detailed mechanism remains unclear, the immune tolerant microenvironment of the liver may induce local and systemic anti-tumor immune suppression upon establishment of metastatic lesions[49–51]. Given that patients with lung-only metastases represent a small subset of patients, with some studies suggesting they comprise as low as 2–3% of metastatic CRC patients, despite promising activity, an active ICI combination regimen has not been identified in patients with CRC with metastasis to liver. The population enrolled in CAMILLA was characterized by higher proportion of patients with liver metastases (79%) as well as higher tumor burden (100% had metastases at 3 or more sites), lower performance status (90% ECOG 1) and progression on multiple prior therapies (52% progressed on 3 or more therapies). Despite these patient characteristics that have been associated with lower ICI response, cabozantinib plus durvalumab demonstrated an ORR of 27.5% (95% CI, 12.73–47.2) including 4 patients with liver metastases with partial responses. Thus, baseline characteristics cannot fully explain the anti-tumor activity of cabozantinib plus durvalumab and supports the notion that cabozantinib potentiates ICI activity in metastatic pMMR/MSS CRC regardless of the presence or absence of liver metastases. While the sample size and the post-hoc nature of the analyses by liver metastases caution against drawing definite conclusions, these findings open the possibility that combined cabozantinib plus durvalumab overcomes the immune

tolerant environment of the liver. Confirmation of these findings in the phase III setting is warranted.

Results from our post-hoc correlative analysis suggest that patients with *RAS* wild-type tumors may have deeper and more durable responses to cabozantinib plus durvalumab. This prompted a post-hoc analysis of the CRC cohort of COSMIC-021 trial which showed that cabozantinib plus atezolizumab (anti-PD-L1) also had more favorable response rates (25% vs. 0%), mPFS (5.8 vs. 2.7 months), and median OS (16.7 vs. 8.7 months) in patients with *RAS* wild type versus mutant pMMR/MSS metastatic CRC[52]. Similarly, higher ORR, mDOR, and mPFS were seen in *RAS* wild-type previously treated pMMR/MSS metastatic CRC patients treated with pembrolizumab and chemotherapy (KEYNOTE-651)[53]. Moreover, the presence of *RAS* alterations was also associated with poor ICI responses in dMMR/MSI-H metastatic CRC patients[54]. Although the exact mechanism in which *RAS* mutations may confer immune resistance is unknown, prior studies in CRC models suggest that hot-spot point mutations in *RAS* promote an immune suppressive TME via facilitation of angiogenesis and recruitment of cancer-associated fibroblasts and myeloid-derived suppressor cells (MDSCs)[55]. Furthermore, in a phase II trial of cabozantinib alone in pMMR/MSS metastatic CRC patients, eleven out of eighteen patients who had achieved a 12-week PFS had *RAS* wild-type tumors. Thus, *RAS* mutations may alternatively have an adverse effect on the treatment response of pMMR/MSS CRC to cabozantinib. Nonetheless, additional studies are warranted to elucidate the downstream pathways responsible for the differential response in *RAS* wild type versus mutant pMMR/MSS metastatic CRC.

Our exploratory GeoMx DSP analysis revealed several transcriptomic characteristics potentially associated with treatment response to cabozantinib plus durvalumab combination therapy. The responders showed evidence of increased angiogenesis through upregulation of VEGF signaling coupled with tumor overexpression of the VEGF receptor 2 (*VEGFR2*) ligand encoding gene *VEGF-A*, suggestive of a potential ligand dependent activation mechanism. Thus, modulation of angiogenesis through VEGF receptor inhibition by cabozantinib is a possible mechanistic explanation. Additionally, we noted the upregulation of MET signaling pathway – another target receptor of cabozantinib – in responders. Of note, the four responders included in the DSP study, had no *MET* gene alteration detected on their tumor comprehensive genomic profiling, which was performed as part of a SOC testing at a CLIA-certified lab before trial enrollment. Given that overexpression of the *MET* gene or hepatocyte growth factor (*HGF*) gene, which encodes a ligand for c-MET, were not observed, the mode of MET signaling pathway activation is likely explained by a mechanism independent of direct *MET or HGF* alterations. Integrin-mediated activation of c-MET has been reported in experimental studies and has been shown to promote invasion and metastasis[56,57]. As integrin receptors lack intrinsic catalytic activity, they depend on other associated proteins to transduce signals initiated by binding to ECM proteins (*i.e.*, fibronectin and osteopontin). Typically, focal adhesion kinase (FAK) and Src family kinases mediate integrin signaling, which influence signal transduction pathways shared with many RTKs (including c-MET), such as PI3K/AKT and RAS-ERK[58,59]. Several mechanisms have been reported to explain joint integrin RTKs interaction and activated signaling. One of these reported mechanisms involves the p loop phosphorylation of RTKs through integrin-mediated activation of Src[59,60]. Given the significant upregulation of cell adhesion signaling, ECM encoding genes and MET signaling pathway upregulation in responders, integrin-mediated activation of c-MET is a potential explanation for MET pathway activation and resultant treatment response to cabozantinib through c-MET inhibition.

Furthermore, the six patients with confirmed partial response including the four in our exploratory GeoMX DSP analysis, all had *RAS/RAF* wild-type tumors and had received and failed EGFR monoclonal

antibody (mAb) prior to trial enrollment. Activated MET as well as integrin signaling have been described as a mechanism of resistance to RTK inhibitor therapy including EGFR mAbs[61–63]. Our results of significantly upregulated MET signaling and cell adhesion signatures in responders may represent either acquired or primary resistance to EGFR mAbs.

Exploratory transcriptomic profiling of the tumor microenvironment in responders showed characteristics of preexisting active anti-tumor immune responses with several counteractive factors including suppressive immune cells (i.e. Tregs) and immune cell infiltration barriers. The presence of features indicative of increased angiogenesis, dense fibrotic TME with increased ECM activity, represent potential migration barriers to activated T cells, preventing their effective interaction with tumor cells[64,65]. Overall, our findings suggest that cabozantinib plus durvalumab clinical synergy is potentially explained by cabozantinib induced modulation of angiogenesis by VEGF receptor inhibition and ECM modulation possibly through interruption of joint integrin c-MET signaling leading to an improved T-cell migration into the tumor core with synergistic durvalumab promoted effective anti-tumor immune response.

When compared to other VEGFR2 TKIs that were previously combined with PD-1/PD-L1 inhibitors in this population, namely regorafenib and lenvatinib, cabozantinib is the only TKI that targets the MET pathway. Our exploratory findings, particularly the observed enrichment of tumor responses in the *RAS* wild-type subgroup along with upregulation of MET signaling pathway in responders, likely explain the more favorable outcome seen in the CAMILLA and COSMIC-021 trials.

This study, despite its merits, has some limitations. Namely, the single-arm design, the overall small sample size, and the exploratory nature of the subgroup analysis caution against overinterpretation of the predictive value of noted molecular findings including the *RAS* mutation status. Additionally, due to the modest sample size of the transcriptomic analyses, subgroup analyses by metastatic site of biopsy was not feasible, especially considering that 70% (14/20) were from the liver. As the site of biopsy will likely have an impact on transcriptomic findings, this represents an important future analysis warranted in larger studies.

In conclusion, cabozantinib plus durvalumab demonstrated promising activity and manageable toxicity in heavily treated patients with pMMR/MSS metastatic CRC. Our post-hoc correlative analyses suggest that *RAS* wild-type status is a potential predictive biomarker in this treatment setting. Additionally, responders to cabozantinib plus durvalumab showed features suggestive of preexisting active anti-tumor immune responses, tumor upregulation of VEGF and MET signaling as well as ECM activity and cell adhesion signatures. This study has led to the ongoing phase III STELLAR-303 randomized trial evaluating XL-092 (cabozantinib analog) plus atezolizumab in patients with pMMR/MSS metastatic CRC in the chemotherapy-refractory setting[66]. Our clinical and exploratory molecular results will be further explored and validated in the STELLAR-303 trial.

## Methods

### Study design and participants
CAMILLA is a single-center, open-label, phase I/II multi-cohort trial evaluating cabozantinib plus durvalumab in patients with advanced, treatment-refractory, gastrointestinal malignancies (NCT03539822). The initial completed phase I part of the study has been published[21]. There are four tumor-specific phase II cohorts of CAMILLA. This is the final report of cohort 2, which consists of patients with unresectable or metastatic CRC. Patients were administered durvalumab 1500 mg intravenously every 28-day cycles and cabozantinib at the recommended phase II dose (RP2D) of 40 mg daily throughout the 28-day cycle. Key study inclusion criteria were the following: (1) advanced or unresectable histologically confirmed colorectal adenocarcinoma; (2)

progression or intolerance to at least two prior SOC systemic therapy including progression on an EGFR antibody if tumor is known *RAS* wild type; and (3) known MSI or MMR status at baseline tumor biopsy. Baseline biopsies were obtained from a metastatic site, which was determined at the discretion of treating investigator. The MSI and MMR testings were only accepted if done in a CLIA-certified lab either locally or through a CLIA-certified commercial vendor. All patients' tumor samples underwent targeted NGS panel testing as part of SOC practice. This testing covers BRAF and extended RAS mutations which include KRAS and NRAS. Patients were excluded if they had prior treatment with an anti-PD-1, -PD-L1, or -PD-L2 agent or monoclonal antibodies targeting MET receptor or HGF or cabozantinib or other tyrosine kinase inhibitors targeting c-MET. This study was approved by the Kansas University Medical Center (KUMC) Institutional Review Board (IRB) (reference number: IRB1#: IRB00000161). All participants provided written consent at the time of enrollment in accordance with the Declaration of Helsinki.

### Procedures and assessments
Details of the phase I part of the trial including the dose limiting toxicities evaluation and subsequent dose-expansion have been published[21]. The pre-planned total duration of treatment was twelve cycles; however, treatment was continued until disease progression, patient request to discontinue treatment, intolerable toxicity, or whichever occurred first. Patients were evaluated every 2 cycles using modified RECIST version 1.1 and were eligible for evaluation of response if at least two tumor imaging scans had been done, namely the baseline pretreatment scan and at least one post-treatment scan at the 8 weeks timepoint. The observed TRAEs were summarized by type and severity according to the Common Terminology Criteria for Adverse Events (CTCAE) version 5.0.

### End points
The primary outcome was ORR defined as the proportion of patients who achieved a complete (CR) or partial response (PR). Confirmed partial response is defined as partial response that is maintained at least 4 weeks apart. Secondary outcome measures were proportion of patients with adverse events as determined by CTCAE version 5.0; DCR as defined by the proportion of patients who achieved CR, PR, or stable disease (SD); PFS as defined by the time from the initiation of treatment until disease progression or death; and OS as defined by the time from initiation of treatment to death.

### Statistical analysis
**Sample size justification.** Based on the Simon's 2-stage Optimum design, a sample size of 29 patients was determined to yield 80.9% power to accept the experimental therapy for further development if the null response rate is 0.04 (or lower) and the target response rate is 0.15 (or higher). The first stage was planned to be assessed after the 23rd patient was assessed for at least 8 weeks; the study proceeded to the second stage to add 6 additional patients if at least 2 responses were noted in the initial 23 patients. Study results were considered to be positive if at least 3 responses were seen in the 29 evaluable patients using an alpha of 0.10. Patients treated in the phase Ib part at the recommended phase II dose were counted in the phase II size calculations. The PASS 2020 software was used to construct the 2-stage Optimum design and the first-stage rejection threshold was fixed at 1 to ensure a 76.6% probability of early termination at stage I of the trial.

**Survival analysis.** Descriptive statistics were reported for baseline patient characteristics. The primary outcome of ORR and the secondary outcome of DCR and 4-month PFS rates were reported as proportions with 95% confidence intervals. Likewise, PFS and OS were reported using median and the corresponding 95% confidence interval. For calculation of median PFS and median OS, all live subjects (or

no progression patients) were flagged as censored on 03/01/2023. Kaplan Meier curves for both PFS and OS were generated, and, for post-hoc analysis, curves stratified by *RAS* wild type and mutant were generated. All statistical analyses (and figures) were conducted using the SAS 9.4 statistical software.

## Transcriptomic digital spatial profiling

Exploratory transcriptomic analysis was performed on tumor baseline biopsies obtained from various metastatic sites before combined cabozantinib and durvalumab treatment. We have chosen ST profiling to investigate the ST characteristics of the tumor and its microenvironment associated with treatment response. Unlike bulk transcriptomics, where the tumor transcriptomic profile is usually influenced by the strong stromal signal that potentially leads to biological misinterpretation, ST provides spatial information and a higher resolution molecular insight into the biological signal derived from the tumor epithelium versus stroma[67]. We utilized the novel ST platform, GeoMx Digital Spatial Profiler (NanoString Technologies; DSP) and the human GeoMx Cancer Transcriptome Atlas (CTA) assay, which measures expression of ~1800 cancer relevant genes. The GeoMx CTA is designed specially to provide a complete coverage map of differential gene expression of the tumor and its microenvironment. The assay uses a cocktail of in situ hybridization (ISH) probes that bind to the mRNA targets within the fixed tissue sections. The ISH probes are conjugated to light-sensitive, photocleavable DNA-barcoded oligos to provide highly multiplexed capacity. Fluorescently labeled antibodies are used to elucidate the cell types of interest that aid in defining regions of interest (ROI) within the tissue for analysis. DSP technology and the GeoMx platform has been extensively used and reviewed since its inception for many studies including colon cancer[68–75].

DSP experiments were performed according to the manufacturer's protocols for manual slide preparation, hybridization, staining, collection, quantification (using NGS based-counting), and data normalization[67]. Briefly, this entailed an initial review of hematoxylin and eosin (H&E)-stained sections by a pathologist to verify the presence of sufficient tumor tissue and to guide subsequent selections of ROI. Next, a 5-μm-thick unstained formalin fixed paraffin embedded section was obtained for each biopsy sample and the slide was prepared for overnight hybridization using a cocktail of the ISH probes. Following overnight hybridization, the slides were stained with morphological markers to identify the tumor epithelium using Alexa Fluor 532 conjugated pan-cytokeratin (pan-CK) antibody (clone AE-1/AE-3, Novus Biologicals) and immune cells using Alexa Fluor 594 conjugated CD45 antibody (clone 2B11 + PD7/26, Novus Biologicals). Syto-13 was used to stain DNA within nuclei. Background liver hepatocytes was identified using Alexa Fluor 532 conjugated cytokeratin 8/18 antibody (clone K8.8 + DC10, Novus Biologicals). The hybridized and fluorescently stained slides were scanned on the GeoMx instrument and ROI were selected to include tumor and stroma interface reflective of TME. An average of 3–4 ROI were selected per sample to capture tumor heterogeneity. The ROI were further segmented into two areas of illumination (AOI) based on the staining pattern of the morphological markers: tumor epithelium (*i.e.*, pan-CK positive cells) and the surrounding peritumoral stroma reflective of TME (*i.e.*, CD45 positive cells plus all other pan-CK negative cells within the selected ROI. (Supplementary Fig. 1 provides an example of ROI selection and segmentation). After ROI selection and segmentation, the photocleavable DNA-barcoded-oligos from each segment were collected into individual wells of a 96-well plate. Once all tissue samples were processed and collected, NGS based quantification was performed. This involved library preparation and assessing the quality and quantity of the purified library using Agilent Bioanalyzer and Qubit assays, respectively. NGS libraries were sequenced using the Illumina NextSeq 550 instrument.

## Transcriptomic data processing and analysis

FASTQ sequencing files were processed using the GeoMx NGS Pipeline (version 2.3.3.10) to generate digital count conversion (DCC) files which were subsequently analyzed using the GeoMx DSP Analysis Suite (version 2.4.1) to perform data quality control (QC) and normalization. QC was performed using the recommended thresholds by the manufacture (NanoString). AOI with QC flags were excluded from downstream analysis. Probes with QC flags were excluded from all AOI. After QC the expression of 1812 genes from 105 AOIs (52 AOIs from tumor epithelium and 53 from stroma) of 20 patients were included in the analysis. The read counts were normalized to the third quartile (Q3) of all selected probes. Normalized gene expression data were then used for further downstream bioinformatics analyses. The normalized counts were downloaded into RStudio. DESeq2 was used for differential expression analysis to identify differentially expressed genes (DEGs) between responders and non-responders in tumor epithelium and stroma, respectively, across all samples[76]. The DESeq2 model was fitted on the gene expression matrix to retrieve the differentially expressed genes between responders and non-responders, by constructing a model matrix design combining the treatment response and patient ID to account for repeated measurements. To ensure biologically meaningful results, in addition to statistical significance we set a cutoff on the fold change (FC) of expression levels at $|\log_2 FC| > 1$ for upregulated genes and $|\log_2 FC| < 1$ for downregulated genes. The clusterProfiler package was used for the gene ontology (GO) analysis to identify enriched biological processes (BP), cellular components (CC) and molecular function (MF) terms in DEGs[77]. Volcano plots and heatmaps were generated using the EnhancedVolcano and pheatmap packages in R (v4.2.2 or later). The scaled Q3-normalized expression of the differentially expressed genes were used in visualizing the heatmaps. All p-values from the differential expression and GO analyses were adjusted for multiple comparisons using the Benjamini−Hochberg corrections with a false discovery rate (FDR) < 0.05. To estimate cell abundance in each AOI, we implemented the SpatialDecon package, which is designed to be used for mixed cell deconvolution in spatial gene expression datasets[75]. The beta values, which represent cellular abundance estimates, were pooled for AOI from each patient using their geometric mean of cell abundance and then the Wilcoxon rank sum test was utilized to compare cell abundance estimates between responders and non-responders. A *p*-value < 0.05 was used to define significant difference. Heatmaps were visualized using the scaled cell abundance (scaled beta values) in the stroma and tumor compartments. All visualization were carried out using ggplot2 package[78]. To investigate the enriched pathways between responders and non-responders, we implemented Gene Set Variation Analysis (GSVA) using 112 annotated gene sets of CTA provided by NanoString as well as the Reactome curated database[79]. The GSVA scores were fitted into a LIMMA linear model fit using the limma package to retrieve the upregulated and downregulated pathways between responders and non-responders using a model matrix constructed for treatment status and patient ID to account for repeated measurements[80]. All p-values from GSVA analyses were adjusted for multiple comparisons using the Benjamini−Hochberg corrections with a false discovery rate (FDR) < 0.05. Additionally, we investigated the enrichment of the 18-gene TIS in the stroma regions between responders and non-responders[29]. The TIS was calculated using the penalized regression model (least absolute shrinkage and selection operator-LASSO)[81]. The Q3 expression of the 18-genes were fitted into LASSO model to retrieve regression model coefficients. Genes with non-zero LASSO coefficients were multiplied by the gene expression to retrieve the weighted gene matrix. The weighted gene sum was used to calculate the final TIS score (Supplementary Data 4). The TIS score was then pooled across patient ID and compared between treatment groups using Wilcoxon rank sum test, and *p*-value < 0.05 to define significant difference.

**Reporting summary**

Further information on research design is available in the Nature Portfolio Reporting Summary linked to this article.

## Data availability

The study protocol is available within the Supplementary Information. The RNA sequencing data have been deposited into NCBI GEO database (accession code: GSE254054). The raw clinical data are available under restricted access due to privacy laws, access to de-identified data can be obtained via written request that specifies the intended use of the data made to the corresponding author. The corresponding author will respond to any request for data sharing within 2 weeks.

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

## Acknowledgements

This trial was funded by research grants from Astrazeneca and Exelixis. The authors would like to thank all the participating patients and their families. We would like to acknowledge support from the University of Kansas (KU) Cancer Center's Biospecimen Repository Core Facility (BRCF) staff for helping obtain human specimens and performing histological work. The BRCF staff was supported in part by a grant from the National Cancer Institute, *i.e.*, the KU Cancer Center's Cancer Center Support Grant (P30 CA168524). We would also like to acknowledge the support of the Kansas Institute for Precision Medicine COBRE Biobanking and Biomarker Validation Core staff (NIGMS P20 GM130423) for support of the DSP studies. Additional acknowledgement to the University of Pittsburgh Medical Center (UPMC) Hillman Cancer Center's Hillman Senior Fellows for Innovative Cancer Research Program.

## Author contributions

Conceptualization, A.S.; methodology, A.S., A.Z.S., M.P., A.K.G. and H.P.; software, J.D., M.P., H.P., and A.N.A.; validation, A.S. and A.Z.S.; formal analysis, J.D., M.P., A.N.A. and A.Z.S.; investigation, A.S., R.P., H.P., A.N.A., J.D., M.P., R.A., A.K., J.B., W.S., S.W., Y.C., H.U.O., R.M., H.A., K.M. (Kelly Mulvaney), A.K.G., A.Z.S.; resources, A.S., M.P., A.K.G. and A.N.A.; data curation, A.S., J.D., M.P., and K.M. (Kelly Mulvaney); writing—original draft preparation, R.P., A.S., A.Z.S., M.P., and H.P.; writing—review and editing, A.S., R.P., H.P., A.N.A., J.D., M.P., R.A., A.K., J.B., W.S., S.W., Y.C., H.U.O., R.M., H.A., K.M. (Kelly Mulvaney), A.K.G., A.Z.S.; visualization, R.P., J.D., A.S. Y.C., H.U.O., and A.Z.S.; supervision, A.S. and A.Z.S.; project administration, A.S.; funding acquisition, A.S. All authors have read and agreed to the published version of the manuscript.

## Competing interests

A.S. reports a leadership role with Autem therapeutics, Exelixis, KAHR medical and Bristol-Myers Squibb; consulting or advisory board role with AstraZeneca, Bristol-Myers Squibb, Merck, Exelixis, Pfizer, Xilio therapeutics, Taiho, Amgen, Autem therapeutics, KAHR medical, and Daiichi Sankyo; institutional research funding from AstraZeneca, Bristol-Myers Squibb, Merck, Clovis, Exelixis, Actuate therapeutics, Incyte Corporation, Daiichi Sankyo, Five prime therapeutics, Amgen, Innovent biologics, Dragonfly therapeutics, Oxford Biotherapeutics, Arcus therapeutics, and KAHR medical; and participation as a data safety monitoring board chair for Arcus therapeutics. R.A. reports research grants (to institution) from AstraZeneca, Bayor, Merck, Bristol Myers Squibb, Exelixis, and Eureka Therapeutics and stock ownership in Actinium Pharmaceuticals and Seagen. A.K. reports research funding (to institution) from Astellas, Tesaro, and Bavarian Nordic. J.B. reports research grants (to institution) from Astellas, Chungchun Intellicrown, Impact Therapeutics, Poseida Therapeutics, Takeda Oncology, and Genome Co. and consultant fees (to institution) from Sanofi. S.W. reports stock/other ownership in Horizon Therapeutics, Iovance Biotherapeutics, and Merus and institutional research funding from Aleon Pharma, Astellas Pharma, Bayer Health, Bristol Myers Squibb, Daiichi Sankyo, EMD Serono, Merck Serono, Nektar, Novartis, Pharmacyclics, AbbVie, Regeneron, Rogosin Institute, Sanofi, Seagen, and Sotio. A.K.G. reports research funding from Predicine and VITRAC Therapeutics, is a co-founder of Sinochips Diagnostics, and serves as a scientific advisory board member to Biovica, Clara Biotech, and Sinochips Diagnostics. The remaining authors report no conflicts of interest.
