## [Peer Review File · Nature Communications]

REVIEWERS' COMMENTS

Reviewer #1 (Remarks to the Author):

Saeed et al. report the results of the MSS/pMMR CRC cohort from the phase II CAMILLA study. The authors also report the results of the post-hoc translational analysis on baseline tissue samples to identify potential predictive biomarkers of response. The clinical trial shows promising results, which will be confirmed in the ongoing phase III trial. Although limited number of patients included in the translational study, results shed new hypothesis into an important clinical need.

Minor comments:

- Abstract, first sentence, include the word patients: "Chemotherapy-refractory proficient mismatch repair/microsatellite stable (pMMR/MSS) metastatic CRC has limited treatment options and PATIENTS usually do not respond to immune checkpoint 36 inhibitor (ICI) therapies"

- Change "&" for "and" in the abstract and throughout the text.

- Line 91: change "great unmet need" to "great unmet clinical need"

- Line 304 "RAS wild type patients" – "change to RAS wild type tumors". Patients are not RAS wt, tumors are. Please review and change all through the manuscript.

- Table 1: Prior lines of therapy: change avastatin for bevacizumab; lonsurf for trifluridine/tipiracil

- Table 2: define abbreviations (PFS, CI) in footnote

- Table 4: please do not use "carbo" or "durba" but the complete name of drugs

Reviewer #3 (Remarks to the Author):

Thank you to the authors for addressing many of the provided comments. While the revised manuscript is improved, my concerns have not been fully addressed. I remain concerned that portions of the manuscript are not aligned with the prespecified analysis plan, and that the post-hoc nature of some analyses is not made transparent.

Below are the comments that have not been adequately addressed:

1. (See original comment 1a) As previously requested, please strike the sentence “RAS wild type status was associated with higher response rates and longer survival” from the abstract (lines 61-62). This comparison was not prespecified nor the main intent of the study. Furthermore, a comparative claim is not justified as the authors did not conduct any statistical tests (statistical testing is likely not appropriate in this case). The author’s post hoc justification for including this statement in the abstract does not have bearing on the prespecified analysis plan of the present study nor does it preclude the need for adequate statistical support.

The presentation of descriptive statistics for the RAS wild type subgroup in lines 53-54 of the abstract is a sufficient presentation of these results.

Thank you to the authors for including the suggested framing of post-hoc analysis in other parts of the manuscript.

2. (See original comment 3) As commented previously, the protocol specifies the 4-month PFS rate (rather than the 6-month PFS rate) as a secondary endpoint (Section 14.3.1). The 4-month PFS rate has been added, but the post-hoc 6-month PFS rate remains and is presented alongside the 4-month PFS rate without adequate framing (lines 52, 177, 299; Table 2). It is misleading to present a post-hoc endpoint alongside the prespecified endpoints without adequate transparency. Please remove the 6-month PFS rate from Table 2 (which summarizes the prespecified efficacy results) and, in text, indicate that this analysis was post hoc.

Similarly, in the Statistical Analysis section, lines 176-177 incorrectly present DCR and PFS as primary outcomes. Please remove the wording “primary outcomes” here.

3. (See original comment 5) As commented previously, although the report has been framed to address an unmet need in metastatic pMMR/MSS colorectal cancer, the study did not prospectively have this intent. While the enrolled cohort ultimately consisted of patients with metastatic pMMR/MSS colorectal cancer, the study inclusion criteria were broader and allowed for patients with unresectable stage 3 disease and dMMR/MSI-H tumors. The study sampled from a broader population and was designed to support conclusions about the broader population.

While this issue was partially addressed, please remove this emphasis on metastatic pMMR/MSS colorectal cancer when summarizing the intent and design of the trial (lines 115, 124) and from the topline conclusion in the abstract (line 61) to better align the report with the prospective intent of the trial.

Reviewer #4 (Remarks to the Author):

Saeed and colleagues present findings from a phase II trial demonstrating a notable response rate in chemotherapy-refractory colorectal cancer patients treated with the combination of cabozantinib and durvalumab. Additionally, they explore biomarkers associated with response using spatial transcriptomics technology on tumor samples, bridging clinical investigation with mechanistic exploration. I commend the authors for this integrative approach, given the complex clinical setting, though I acknowledge the challenges of identifying clear correlates of response given the limited sample set and limited clinical responses, a point that might benefit from acknowledgment in the discussion.

However, caution is warranted in interpreting the results. The authors assert strong claims based on a limited sample set analyzed by spatial transcriptomics, and these conclusions should be nuanced accordingly. Some observations, such as increased gene expression and activity in pathways like EMT, appear difficult to reconcile with improved responses to immunotherapy. Moreover, while deconvolution approaches are suitable for discovery, validation through techniques establishing a ground truth is ideal. Are there available slides for validation, particularly regarding the increased infiltration of memory CD8+ T cells in responders?

Given that responders were exclusively KRAS wild-type, the spatial transcriptomic comparison for non-responders should include only KRAS wild-type samples, as KRAS status likely impacts downstream signaling. The presentation of results could be clarified, and the quality of figures improved. For instance, the necessity of Figure 7 as a standalone should be reconsidered.

Additional comments:

1. The term "meaningful" in "cabozantinib plus durvalumab demonstrated meaningful activity" is subjective; specificity would enhance clarity.
2. The generalization that MMR-p CRCs "have non-inflamed TME" is inaccurate; inflammatory features exist, though not necessarily associated with Th1-oriented, cytotoxic anti-tumor immune responses.
3. The mention of "complete response" in the Efficacy section is confusing, especially if no complete responses were observed according to Figure 2's legend; clarification is needed.
4. The assumption that KRAS status significantly influences immunotherapy responses should be supported by evidence in the discussion, addressing whether KRAS status affects cabozantinib activity.
5. Clarify how fibroblasts are enriched in TEC if TEC was selected based on keratin; consider the difficulty in distinguishing EMT processes from fibroblast-associated molecules in transcriptomic approaches.
6. Correct the reference for TIS validation, as Reference 36 is not Ayers.

REVIEWERS' COMMENTS

Reviewer #1 (Remarks to the Author):

Saeed et al. report the results of the MSS/pMMR CRC cohort from the phase II CAMILLA study. The authors also report the results of the post-hoc translational analysis on baseline tissue samples to identify potential predictive biomarkers of response. The clinical trial shows promising results, which will be confirmed in the ongoing phase III trial. Although limited number of patients included in the translational study, results shed new hypothesis into an important clinical need.

Minor comments:

- Abstract, first sentence, include the word patients: "Chemotherapy-refractory proficient mismatch repair/microsatellite stable (pMMR/MSS) metastatic CRC has limited treatment options and PATIENTS usually do not respond to immune checkpoint 36 inhibitor (ICI) therapies"

Reply: Authors appreciate the comments. The abstract has been summarized per the journal guidelines and the word "patients" has been added where applicable.

- Change "&" for "and" in the abstract and throughout the text.

Reply: Authors appreciate the comments, which have been addressed.

- Line 91: change "great unmet need" to "great unmet clinical need"

Reply: Authors appreciate the comments, which have been addressed.

- Line 304 "RAS wild type patients" – "change to RAS wild type tumors". Patients are not RAS wt, tumors are. Please review and change all through the manuscript.

Reply: Authors appreciate the comments, which have been addressed.

- Table 1: Prior lines of therapy: change avastatin for bevacizumab; lonsurf for trifluridine/tipiracil

Reply: Authors appreciate the comments, which have been addressed. Respective words have been changed.

- Table 2: define abbreviations (PFS, CI) in footnote

Reply: Authors appreciate the comments, which have been addressed. Footnote has been revised.

- Table 4: please do not use "carbo" or "durba" but the complete name of drugs

Reply: Authors appreciate the comments, which have been addressed. Cabo and Durva have been revised to cabozantinib and durvalumab throughout the manuscript.

Reviewer #3 (Remarks to the Author):

Thank you to the authors for addressing many of the provided comments. While the revised manuscript is improved, my concerns have not been fully addressed. I remain concerned that portions of the manuscript are not aligned with the prespecified analysis plan, and that the post-hoc nature of some analyses is not made transparent.

Below are the comments that have not been adequately addressed:

1. (See original comment 1a) As previously requested, please strike the sentence “RAS wild type status was associated with higher response rates and longer survival” from the abstract (lines 61-62). This comparison was not prespecified nor the main intent of the study. Furthermore, a comparative claim is not justified as the authors did not conduct any statistical tests (statistical testing is likely not appropriate in this case). The author’s post hoc justification for including this statement in the abstract does not have bearing on the prespecified analysis plan of the present study nor does it preclude the need for adequate statistical support.

The presentation of descriptive statistics for the RAS wild type subgroup in lines 53-54 of the abstract is a sufficient presentation of these results.

Thank you to the authors for including the suggested framing of post-hoc analysis in other parts of the manuscript.

Reply: Authors appreciate the comments, which have been addressed. The statement “RAS wild type status was associated with higher response rates and longer survival” has been removed.

2. (See original comment 3) As commented previously, the protocol specifies the 4-month PFS rate (rather than the 6-month PFS rate) as a secondary endpoint (Section 14.3.1). The 4-month PFS rate has been added, but the post-hoc 6-month PFS rate remains and is presented alongside the 4-month PFS rate without adequate framing (lines 52, 177, 299; Table 2). It is misleading to present a post-hoc endpoint alongside the prespecified endpoints without adequate transparency. Please remove the 6-month PFS rate from Table 2 (which summarizes the prespecified efficacy results) and, in text, indicate that this analysis was post hoc.

Reply: Authors appreciate the comments, which have been addressed. 6 month PFS has been removed from Table 2 and the statement has been changed to “4-month PFS rate was 44.83%; and the 6 month PFS, which was assessed as a post-hoc outcome, was 31%.”

Similarly, in the Statistical Analysis section, lines 176-177 incorrectly present DCR and PFS as primary outcomes. Please remove the wording “primary outcomes” here.

Reply: Authors appreciate the comments, which have been addressed. “The primary outcome of ORR and the secondary outcomes of DCR and 4-month PFS rates were reported as proportions with 95% confidence intervals”.

3. (See original comment 5) As commented previously, although the report has been framed to address an unmet need in metastatic pMMR/MSS colorectal cancer, the study did not prospectively have this intent. While the enrolled cohort ultimately consisted of patients with metastatic pMMR/MSS colorectal cancer, the study inclusion criteria were broader and allowed for patients with unresectable stage 3 disease and dMMR/MSI-H tumors. The study sampled from a broader population and was designed to

support conclusions about the broader population.

While this issue was partially addressed, please remove this emphasis on metastatic pMMR/MSS colorectal cancer when summarizing the intent and design of the trial (lines 115, 124) and from the topline conclusion in the abstract (line 61) to better align the report with the prospective intent of the trial.

Reply: Authors appreciate the comments, which have been addressed. The wording has been revised.

Reviewer #4 (Remarks to the Author):

Saeed and colleagues present findings from a phase II trial demonstrating a notable response rate in chemotherapy-refractory colorectal cancer patients treated with the combination of cabozantinib and durvalumab. Additionally, they explore biomarkers associated with response using spatial transcriptomics technology on tumor samples, bridging clinical investigation with mechanistic exploration. I commend the authors for this integrative approach, given the complex clinical setting, though I acknowledge the challenges of identifying clear correlates of response given the limited sample set and limited clinical responses, a point that might benefit from acknowledgment in the discussion.

However, caution is warranted in interpreting the results. The authors assert strong claims based on a limited sample set analyzed by spatial transcriptomics, and these conclusions should be nuanced accordingly. Some observations, such as increased gene expression and activity in pathways like EMT, appear difficult to reconcile with improved responses to immunotherapy. Moreover, while deconvolution approaches are suitable for discovery, validation through techniques establishing a ground truth is ideal. Are there available slides for validation, particularly regarding the increased infiltration of memory CD8+ T cells in responders?

Given that responders were exclusively KRAS wild-type, the spatial transcriptomic comparison for non-responders should include only KRAS wild-type samples, as KRAS status likely impacts downstream signaling. The presentation of results could be clarified, and the quality of figures improved. For instance, the necessity of Figure 7 as a standalone should be reconsidered.

Reply:

Thank you for your valuable feedback and comments. We acknowledge the limitations in the current study given the limited sample size and clinical response as well as the exploratory nature of the correlative analysis. More conclusionary statements in the results and discussion sections have been nuanced to reflect such limitations.

Additionally, we also acknowledge the limitations in deconvolution approaches in establishing ground truth regarding the abundance of immune cells. Ideally, the findings from these approaches need to be validated through a gold standard method like immunohistochemistry or immunofluorescence. Given the funding limitations, which was only sufficient to conduct the GeoMx experiment, additional studies couldn't be performed or included in the experimental design. However, we are seeking additional funding to conduct a future study that will validate and build further on the findings from the current GeoMx analysis.

We appreciate your feedback regarding conducting the comparative analysis for the KRAS wild type subgroup. Given the small sample size we took a general approach to identify transcriptomic features associated with clinical response in responders versus non-responders. To address your suggestion, we have performed additional analysis for the RAS wild type subgroup comparing responders to non-responders and the findings are in line with the general analysis. We have highlighted the results from this additional analysis in the results section of the manuscript. The detailed analysis with figures & tables for the RAS wild type subgroup has been included as a supplement to the manuscript.

Thank you for your suggestions regarding the figures. The figures have been edited and figure 7 has been splitted into two figures.

Additional comments:

1. The term "meaningful" in "cabozantinib plus durvalumab demonstrated meaningful activity" is subjective; specificity would enhance clarity.

Reply: Authors appreciate the comments, which have been addressed. "Meaningful activity" has been revised to "demonstrated anti-tumor activity".

2. The generalization that MMR-p CRCs "have non-inflamed TME" is inaccurate; inflammatory features exist, though not necessarily associated with Th1-oriented, cytotoxic anti-tumor immune responses.

Reply: Authors appreciate the comments, which have been addressed. The wording, "non-inflamed" have been removed or revised throughout the manuscript. For example, the above have been addressed: "...have features of TME that are not associated with cytotoxic anti-tumor immune responses..." and "...modulating the TME to a state that is associated with cytotoxic anti-tumor immune responses with..."

3. The mention of "complete response" in the Efficacy section is confusing, especially if no complete responses were observed according to Figure 2's legend; clarification is needed.

Reply: Authors appreciate the comments, which have been addressed. Mention of complete response in the Efficacy Section has been removed.

4. The assumption that KRAS status significantly influences immunotherapy responses should be supported by evidence in the discussion, addressing whether KRAS status affects cabozantinib activity.

Reply: The authors appreciate the comments of the reviewer. The authors respectfully point out that the post-hoc analyses of the association of RAS status and prognosis in our CAMILLA study and COSMIC-021 as well as KEYNOTE-651 and KEYNOTE-177 represent clinical evidence for the hypothesis-generating statement that RAS status may influence immunotherapy responses. Potential rationale may be that RAS mutations are associated with an immune suppressive TME characterized by angiogenesis and recruitment of MDSCs and CAFs as evidenced by Liao et al., referenced in the discussion. We have added an additional reference in the discussion to allude to the possibility of RAS status altering the response to cabozantinib, rather than the immune checkpoint inhibitor - "Furthermore, in a phase II trial of cabozantinib alone in pMMR/MSS metastatic CRC patients, eleven out of eighteen patients who had achieved a 12-week PFS had RAS wild type tumors. Thus, RAS mutations may alternatively have an adverse effect on the treatment response of pMMR/MSS CRC to

cabozantinib". The authors agree that the above results are hypothesis-generating and warrant validation in the phase III setting, which is currently under way in the phase III STELLAR-303 trial.

5. Clarify how fibroblasts are enriched in TEC if TEC was selected based on keratin; consider the difficulty in distinguishing EMT processes from fibroblast-associated molecules in transcriptomic approaches.

Reply:

Thank you for your valuable question and feedback. The experimental design of the DSP study utilized a masking technique that captured oligos from the keratin+ segments (tumor mask) and we didn't include any specific immunofluorescence marker and masking filter to capture immune cells or stromal cells separately. To capture the stromal and immune cells we used an inverse filter (panCK negative). Thus, the tumor segment mask captured intraepithelial immune cells and any entrapped fibroblast nuclei within this area. Given that the tumor cells are the predominant cell type in the keratin + segment, the transcriptomic signal from this segment would be most reflective of the tumor and thus the EMT signature.

Additionally, the SpatialDecon algorithm performed in our study for the deconvolution experiment, utilized a cell profile matrix for the immune and stromal cells characterized based on minimal expression in cancer cell and characterized based on flow-sorted PBMCs, flow sorted stromal cells and single cell RNA-seq of tumors. The profile matrix has 8 fibroblasts genes (COL1A1, COL3A1, COL6A1, COL6A2, DCN, GREM1, PAMR1, TAGLN) that don't overlap with the EMT signature genes utilized in the GSVA analysis. Based on this, the cell abundance score of fibroblasts in tumor segment would be most reflective of signals from the fibroblasts. However, we cannot rule out completely that the abundance prediction is not confounded by tumor intrinsic expression of these fibroblast genes.

6. Correct the reference for TIS validation, as Reference 36 is not Ayers.

Reply:

Thank you for pointing this. This was corrected.